# OpenReview forum: "Right Time to Learn: Promoting Generalization via Bio-inspired Spacing Effect in Knowledge Distillation"
_ICLR.cc/2025/Conference — Submitted to ICLR 2025_

### Official Review · Reviewer_v8iU · 2024-10-30

**Soundness:** 3
**Presentation:** 3
**Contribution:** 2
**Rating:** 8
**Confidence:** 4

**Summary:**

This paper introduces Spaced Knowledge Distillation (Spaced KD), a bio-inspired approach to knowledge distillation that incorporates spacing intervals in the training process to enhance model generalization. Inspired by the "spacing effect" observed in biological learning, Spaced KD involves training a teacher model a few steps ahead of the student model and periodically transferring knowledge at intervals. This technique allows the student model to find flatter minima in the loss landscape, which leads to better generalization and improved robustness to noise. Extensive experiments demonstrate that Spaced KD yields superior performance across multiple datasets and architectures compared to traditional KD methods without additional training costs.

**Strengths:**

1) Novelty: The paper introduces a fresh approach by integrating the bio-inspired “spacing effect” into knowledge distillation (KD), which is relatively unexplored in existing literature.

2) Strong Theoretical Foundation: The authors provide a well-structured theoretical analysis linking Spaced KD to flat minima, supporting claims about enhanced generalization. The use of Hessian trace analysis to demonstrate why Spaced KD leads to better generalization adds rigor and robustness to the claims.

3) Comprehensive Experiments: The experimental results are extensive, spanning multiple datasets (CIFAR-100, Tiny-ImageNet, ImageNet-100, and ImageNet-1K) and network architectures (ResNet variants, DeiT-Tiny, PiT-Tiny). These experiments show consistent improvements with Spaced KD, reinforcing the method’s general applicability.

4) Practicality: Spaced KD is straightforward to implement, making it accessible for a broad range of applications without requiring additional training time or major architectural modifications. This ease of integration enhances its appeal for both academia and industry.

5) Robustness and Generalization Gains: Spaced KD demonstrates superior robustness to noise and out-of-distribution data, an increasingly valued property in modern AI. This robustness adds significant value, especially for real-world applications where models encounter varied data distributions.

**Weaknesses:**

1) Incremental Novelty: While Spaced KD introduces a creative application of the spacing effect to knowledge distillation, it lacks a fundamentally new mechanism within the KD framework. This incremental contribution may limit its overall impact, as the method primarily builds on existing KD techniques without significantly advancing the underlying theory.

2) Interval Sensitivity and Adaptability: The success of Spaced KD hinges on selecting an effective interval for knowledge transfer. However, the paper does not explore adaptive strategies for dynamically tuning this interval, which may constrain Spaced KD’s versatility across diverse tasks and datasets with varying training schedules.

3) Applicability Across Model Sizes and Architectures: Spaced KD is demonstrated only with identical teacher and student models of the same size, without addressing model compression. It remains uncertain how this approach would perform if applied to different model sizes or architectures, such as a transformer teacher model paired with a CNN student. The potential to generalize Spaced KD for cross-architecture or cross-size distillation remains unexplored.

**Questions:**

1) Applicability Across Model Sizes and Architectures: How does Spaced KD perform when the teacher and student differ in size or architecture? Specifically, can it support scenarios where the teacher is larger or has a different architecture (e.g., a transformer teacher with a CNN student), or does it rely on identical model sizes?

2) Comparison with Gradient-based Methods: How does Spaced KD differ from traditional gradient-based methods where gradients are updated after each batch? Given that Spaced KD introduces intervals between teacher and student updates, is this approach essentially a modified gradient update, or does it provide additional benefits that distinguish it from standard gradient optimization?

---

> ### Author Response · Authors · 2024-11-19
> **Author Response**
>
> Thank you for your valuable comments. Below, we provide a point-to-point response to each of the weaknesses (**W**) and questions (**Q**), and summarize the corresponding revisions in the final version. We hope you may consider this a sufficient reason to raise the score. If you have any further questions, please let us know.
>
>
> ### **W1: New fundamental mechanism within the KD framework.**
>
> As recognized in your Strength 1, the spacing effect in KD is ``relatively unexplored in existing literature''. In fact, our work reveals an innovative mechanism that the spacing effect in KD improves generalization of the student model via encouraging the convergence to a flat loss landscape. The innovation is reflected in two aspects:
>
> (1) Unlike previous understandings [1,2] that attribute the efficacy of KD to the knowledge capacity gap between the teacher and the student (where a spacing-like effect should be more effective in the early stage of training), our Spaced KD works in the later stage of training (see Fig.3) around the convergence point (see Sec.4.2). In Appendix A.5, we devise a naive baseline that keeps constant $s$ training steps between the teacher model and the student model, which exhibits no significant improvement over online KD. This suggests that our Spaced KD mainly affects the convergence process of the student model by providing informative directions from the stable teacher model after training $s$ steps.
>
> (2) As shown in a pioneering theoretical analysis [3], KD shares a similar mechanism as ensemble learning in improving generalization. However, current efforts have focused on better ensembling separately trained teacher and student models from a spatial scale. Although online KD and self KD attempt to ensemble jointly trained teacher and student models from a temporal scale, the effect of temporal interval remains under-explored. Our work demonstrates that the implementation of the spacing effect in KD is highly non-trivial (see the first aspect above), and a specially designed temporal interval provides great improvements to KD, applicable to various architectures and datasets.
>
> > [1] Knowledge Distillation and Student-Teacher Learning for Visual Intelligence: A Review and New Outlooks, TPAMI, 2022.
>
> > [2] Knowledge Distillation: A Survey, IJCV, 2021.
>
> > [3] Towards Understanding Ensemble, Knowledge Distillation and Self-Distillation in Deep Learning. ICLR (Outstanding Paper Nomination), 2023.
>
> ### **W2: Interval sensitivity and adaptability.**
>
> As shown in Fig.2, $s$ is relatively stable and a range of $s$ can provide significant improvements, with $s=1.5$ having comparably the strongest performance. Therefore, we set $s=1.5$ for all experiments if not specified, making it an easy-to-use strategy in applications. We agree that the adaptive selection of $s$ is a promising future work, and we have discussed this in our original submission (see Sec.6).
>
>
> ### **W3 & Q1: Applicability across model sizes and architectures.**
>
> Following your suggestion, we conduct a series of experiments with large-size teachers and small-size students, using similar or different architectures. As shown in the following tables, our Spaced KD clearly outperforms the online KD across model sizes and architectures. We have added more explanations (lines 362-364) and empirical results (Tab.7 of Appendix A.4) in our revised manuscript.
>
> Results of large-size teacher with small-size student:
>
> | Student   | Teacher       | Online KD | Spaced KD | Increase |
> | --------- | ------------- | --------- | --------- | -------- |
> | ResNet-18 | ResNet-18 * 1 | 71.05     | 72.43     | **1.38** |
> | ResNet-18 | ResNet-18 * 2 | 71.77     | 72.77     | **1.00** |
> | ResNet-18 | ResNet-18 * 4 | 72.17     | 73.11     | **0.94** |
> | ResNet-18 | ResNet-18 * 8 | 72.03     | 73.52     | **1.49** |
> | ResNet-18 | ResNet-50     | 72.18     | 73.49     | **1.31** |
> | ResNet-18 | ResNet-101    | 71.61     | 73.04     | **1.43** |
>
> Results of transformer teacher with ResNet student:
>
> | Student   | Teacher       | Online KD | Spaced KD | Increase |
> | --------- | ------------- | --------- | --------- | -------- |
> | ResNet-18 | DeiT-Tiny     | 78.61     | 79.38     | **0.77** |
> | ResNet-18 | PiT-Tiny      | 77.13     | 78.77     | **1.64** |
>
>
> ### **Q2: Difference between Spaced KD and traditional gradient-based methods.**
>
> As shown in our theoretical analysis in Sec.4.2, our Spaced KD serves as a well-designed regularization to improve traditional gradient-based methods such as SGD. In particular, unlike the stochastic directions of gradient descent in SGD, our Spaced KD encourages the student model to converge to a more generalizable loss landscape by providing stable informative directions from the teacher model.

---

> > ### Comment · Reviewer_v8iU · 2024-11-26
> >
> > Thank you for the detailed rebuttal and the additional experiments. I appreciate the effort you put into addressing my concerns and providing further evidence. Below are my updated thoughts and two follow-up questions:
> >
> > ---
> >
> > #### **1. Incremental Novelty (W1)**
> > Your clarification on the innovative aspects of Spaced KD, particularly its focus on guiding the student model during the later training stages and the connection to temporal interval ensembling, is appreciated. The theoretical grounding and comparisons with naive baselines add robustness to your claims.
> >
> > ---
> >
> > #### **2. Applicability Across Model Sizes and Architectures (W3 & Q1)**
> > The new experiments involving large-size teachers, small-size students, and cross-architecture distillation are highly valuable. Including these results in the manuscript (e.g., Tab. 7) significantly strengthens your argument for generalizability.
> >
> > ---
> >
> > ### **Follow-up Questions**
> >
> > 1. **Interval Sensitivity:** Could you provide insights into whether different datasets or training schedules (e.g., warm restarts, adaptive learning rates) affect the optimal interval selection? While you show stability across intervals, adaptability could clarify the practicality of Spaced KD in diverse settings.
> >
> > 2. **Teacher Model Stability:** Since Spaced KD relies on the teacher's stability, have you explored its efficacy when the teacher is fine-tuned, noisy, or pre-trained on a domain different from the student’s? This would further demonstrate the method’s robustness in real-world scenarios.
> >
> > ---
> >
> > Your rebuttal addresses most of my concerns effectively, and the additional experimental results further substantiate your claims. These follow-up questions aim to explore Spaced KD’s robustness and adaptability further, which could enhance its applicability and impact.

---

> > > ### Author Response · Authors · 2024-11-26
> > > **Response to Follow-up Questions of v8iU**
> > >
> > > We are pleased that our rebuttal has addressed most of your concerns. Below are our detailed responses to the additional questions:
> > >
> > > ### **Follow-up Question 1: Interval Sensitivity**
> > >
> > > Our study has analyzed the stability of the spacing interval in Section 5.3. Given the dataset and architecture, we follow effective configurations (such as learning rates, batch sizes, and training epochs) from prior work and then analyze the optimal spacing interval. The experimental results show that the optimal spacing interval remains relatively stable across different datasets and architectures. Since the changes in such effective configurations may negatively affect the baseline KD performance, we did not explore their impact on the optimal spacing interval. However, as shown in the analysis of “Learning Rate and Batch Size”, using the default spacing interval (i.e., s=1.5) achieves consistently strong improvements across a range of learning rates and batch sizes, suggesting its stability in these scenarios.
> > >
> > > ### **Follow-up Question 2: Teacher Model Stability**
> > > The primary focus of our work is to improve generalization of the same neural network model without using additional training data. Therefore, we apply the proposed Spaced KD in scenarios where the teacher and student models share the same dataset and architecture, using the same parameter initialization throughout the manuscript. We also discuss the feasibility of applying distillation across different architectures and model sizes, with the results reinforcing the robustness of Spaced KD. We fully agree with your point that examining the effects of teacher-student model differences, particularly across varying data domains, training methods, and architectures, represents an important and valuable area for future research.
> > >
> > > We sincerely appreciate your positive feedback and valuable comments. We hope our rebuttal, revised manuscript, and the current responses may receive more positive ratings from you. Please feel free to contact us if you have any further questions.

---

> ### Author Response · Authors · 2024-11-24
>
> Dear Reviewer v8iU,
>
> If you have any further feedback or concerns, please feel free to let us know. Your input is highly appreciated, and we look forward to hearing from you.
> Thank you once again for your time and consideration.
>
> Best wishes,
>
> Authors

---

> ### Comment · Reviewer_v8iU · 2024-11-30
> **Revised Review and Updated Score**
>
> Dear Authors,
>
> Thank you once again for your efforts in providing additional details, explanations, and experiments to enhance your paper. I have carefully reviewed the comments from the other reviewers as well.
>
> Taking into account all the reviews, feedback, and your thorough rebuttal, I find this work to be both interesting and beneficial to the community. While some of the improvements may not be highly significant, I still believe the contribution is valuable and worthy of recognition. Accordingly, I would like to raise my score to 8.

---

> > ### Author Response · Authors · 2024-12-03
> > **Thank you**
> >
> > We thank the reviewer for finding our response satisfactory and are happy to know the very positive rating.
> >
> > Best wishes,
> >
> > Authors

---

### Official Review · Reviewer_bFLi · 2024-11-03

**Soundness:** 3
**Presentation:** 3
**Contribution:** 3
**Rating:** 6
**Confidence:** 3

**Summary:**

This paper studies a variant of Knowledge Distillation, designed with Space Interval learning, inspired by the spacing effects observed in biological learning.

Design: The Space Interval method is applied to Online KD, where the teacher model is trained a few intervals (s iterations) ahead of the student. The teacher’s weights are then frozen and used to train the student, who lags behind. The hypothesis is that, by updating the teacher less frequently, it provides more stable guidance to the student.

Theoretical Analysis: The authors use the trace of the Hessian matrix as a measure of the loss landscape’s flatness, where a smaller Hessian trace indicates a flatter, and thus potentially more generalizable, landscape. They show that the loss landscape of the new paradigm has flatter Hessian trace ompared to the traditional Online KD, suggesting improved generalization.

Experimental Results: The proposed method outperforms the traditional Online KD on CIFAR-100, Tiny-ImageNet, ImageNet-100, and ImageNet-1K datasets by approximately 1–2% on average with three ResNet variants. The authors also provide experiments demonstrating robustness improvements over online and self-KD counterparts, along with ablation studies on learning rate, batch size, and loss functions.

**Strengths:**

Overall, the paper is easy to follow. The idea seems simple to implement and appears interesting as inspired by biological learning processes. Empirically, it achieves a few percentage points of improvement over existing online methods, as demonstrated in the experimental sections.

**Weaknesses:**

The results are overall encouraging, there are few points to be clarified with the authors in term of bio-inspiration, the generality of the method, the settings used across experiments and the seemingly conflicts between theoretical and empirical results as in Questions section.

**Questions:**

1. While the bio-inspiration is interesting, the learning mechanism applied here differs from the spacing effects observed in biology. In biological learning, spacing effects enhance learning and memory retention of a subject when study sessions or practice trials are spread across intervals of time. However, in this application, the teacher is trained at intervals while the student learns continuously which seems the design is not well-aligned with the biological inspiration. Could the authors provide deeper details on this point to show the bio-inspiration is valid.

2. In the online setting, most experiments use a teacher with the same architecture as the student. To demonstrate the generality of the method, could the authors conducting experiments where the teacher’s architecture is significantly larger than the student's?

3. As shown in Fig. 3, the period of applying Space KD is critical. Are all experiments using the best setting or different experiments has different periods? Could the authors provide details on the optimal settings used across the experiments in the paper and explain how these settings were selected?

4. Is there a conflict between the theoretical analysis and the empirical results? While the theoretical analysis is independent of the period during which Space KD is applied, the empirical results indicate that it is only useful in the last 10 epochs of training? Could the authors provides also the only online KD baseline's results in Fig. 3 for the comparision?

---

> ### Author Response · Authors · 2024-11-19
> **Author Response**
>
> Thank you for your valuable comments. Below, we provide a point-to-point response to each of the weaknesses (**W**), questions (**Q**), and limitations (**L**), and summarize the corresponding revisions. We hope you may consider this a sufficient reason to raise the score. If you have any further questions, please let us know.
>
>
>
> ### **Q1: Connection between the proposed Spaced KD and its bio-inspiration.**
>
> Our proposed Spaced KD for machine learning is inspired by the underlying computational principle of the spacing effect in biological learning:
>
> In machine learning, KD aims to optimize the parameters of a student network with the help of a teacher network by regularizing their outputs to be consistent in response to similar inputs. As shown in a pioneering theoretical analysis [1], KD shares a similar mechanism as ensemble learning (EL) in improving generalization from the training set to the test set. In particular, online KD performs this mechanism at temporal scales, and self KD can be seen as a special case of online KD. In comparison, the biological spacing effect can also be generalized to a kind of EL at temporal scales, as the brain network processes similar inputs with a certain time interval and updates its synaptic weights based on previous synaptic weights, which allows for stronger learning performance at test time [2-4].
>
> The proposed Spaced KD draws inspiration from the biological spacing effect and capitalizes on the underlying connections between (online) KD and EL. It incorporates a space interval between the student network and the teacher network to improve the performance of generalization, validated through extensive empirical and theoretical analyses. We have added more explanations to make it clearer (see Sec.3.2, lines 193-207).
>
> > [1] Towards Understanding Ensemble, Knowledge Distillation and Self-Distillation in Deep Learning. ICLR (Outstanding Paper Nomination), 2023.
>
> > [2] The Phosphatase SHP2 Regulates the Spacing Effect for Long-Term Memory Induction. Cell, 2009.
>
> > [3] It’s All About Timing. Cell, 2009.
>
> > [4] The Right Time to Learn: Mechanisms and Optimization of Spaced Learning. Nature Reviews Neuroscience, 2016.
>
>
> ### **Q2: Experiments where the teacher's architecture is significantly larger than the student's.**
>
> Following your suggestion, we conduct a series of experiments where the teacher's architecture is significantly larger than the student's, including larger width (from ResNet-18 * 1 to ResNet-18 * 8) and larger depth (from ResNet-18 * 1 to ResNet-101* 1). As shown in the following table, the proposed Spaced KD consistently outperforms the online KD, validating its effectiveness and generality. We have added it in our revised manuscript (see lines 362-364 and Tab.7 in Appendix A.4)
>
> Performance of online KD and Spaced KD on CIFAR-100:
> | Student   | Teacher       | Online KD | Spaced KD | Increase |
> | --------- | ------------- | --------- | --------- | -------- |
> | ResNet-18 | ResNet-18 * 1 | 71.05     | 72.43     | **1.38** |
> | ResNet-18 | ResNet-18 * 2 | 71.77     | 72.77     | **1.00** |
> | ResNet-18 | ResNet-18 * 4 | 72.17     | 73.11     | **0.94** |
> | ResNet-18 | ResNet-18 * 8 | 72.03     | 73.52     | **1.49** |
> | ResNet-18 | ResNet-50     | 72.18     | 73.49     | **1.31** |
> | ResNet-18 | ResNet-101    | 71.61     | 73.04     | **1.43** |
> | ResNet-18 | DeiT-Tiny     | 78.61     | 79.38     | **0.77** |
> | ResNet-18 | PiT-Tiny      | 77.13     | 78.77     | **1.64** |
>
>
> ### **Q3: Experimental setting in Fig.3. How these optimal settings were selected?**
>
> We use CIFAR-100 and $s=1.5$ for all experiments in Fig.3. This is because, in our empirical investigation, we have validated that $s$ is relatively stable and a range of $s$ can provide significant improvements (see Fig.2), with $s=1.5$ having comparably the strongest performance. Therefore, for ease of implementation and fairness of comparison, we set $s=1.5$ in all experiments if not otherwise specified (see lines 411-413). This property reduces the effort of hyperparameter tuning and further enhances the applicability of our Spaced KD.
>
> ### **Q4a: Consistency between the theoretical analysis and the empirical results in Fig.3.**
>
> We respectfully argue that the theoretical analysis and the empirical results are consistent. Our theoretical analysis assumes that the network parameters $\theta$ are close to their convergence point $\theta^*$, rather than far away (see lines 265 and Theorem 4.4). The empirical results in Fig.3 highlight that the proposed spacing effect exerts its greatest influence during the later stage of training which is close to the convergence point.
>
> ### **Q4b: Online KD baseline in Fig.3.**
>
> We have added it in Fig.3, where our Spaced KD provides great improvements on it.

---

> ### Author Response · Authors · 2024-11-24
>
> Dear Reviewer bFLi,
>
> If you have any further feedback or concerns, please feel free to let us know. Your input is highly appreciated, and we look forward to hearing from you.
> Thank you once again for your time and consideration.
>
> Best wishes,
>
> Authors

---

> > ### Author Response · Authors · 2024-11-30
> > **Looking forward to your feedback**
> >
> > Dear Reviewer bFLi,
> >
> > We sincerely appreciate your valuable comments. We have tried our best to provide a point-to-point response and revise our manuscript accordingly. We hope that our extended experiments, explanations and analyses have addressed your concerns for more positive rating. Since the rebuttal period is coming to an end, we would appreciate further feedback from you. Please feel free to contact us if you have any further questions.
> >
> > Authors

---

### Official Review · Reviewer_5J6i · 2024-11-03

**Soundness:** 3
**Presentation:** 3
**Contribution:** 3
**Rating:** 5
**Confidence:** 4

**Summary:**

The paper introduces a new method for online knowledge distillation that uses the teacher model trained ahead of the student at each epoch, resulting in a flatter loss landscape.

**Strengths:**

The paper is well-written and easy to follow.

The experiments show the effective improvement of the proposed method over vanilla KD.

**Weaknesses:**

1) I did not get the connection between the proposed method and the spacing effect in biological learning mentioned in Section 3.2. The teacher has different weights from the student. Why does training for several additional epochs cause a spacing effect?

2) The theoretical part is not robust. The proposed method assumes that the teacher trained ahead is close to the local minimizer $\phi^*$, so why not use a well-trained model instead?

3) There is a lack of comparison with other KD methods like [1,2]

[1] Self-Distillation from the Last Mini-Batch for Consistency Regularization.

[2] Self-Knowledge Distillation with Progressive Refinement of Targets.

**Questions:**

1) What are the numerical values of x and y in Figure 2a?

2) As 0.5 epochs mean training on half of the dataset, would this cause a bias towards that half of the dataset? Is the training dataset shuffled?

3) How can the proposed method be applied to self-distillation in [3], where the teacher is the deepest layer of models and the students are the shallow layers?

4) What dataset was used for Figure 3, and which settings were used for the experiments?



[3] Be Your Own Teacher: Improve the Performance of Convolutional Neural Networks via Self Distillation.

---

> ### Author Response · Authors · 2024-11-19
> **Author Response (Part 1)**
>
> Thank you for your valuable comments. Below, we provide a point-to-point response to each weaknesses comment (**W**), and summarize the corresponding revisions. We hope you may consider this a sufficient reason to raise the score. If you have any further questions, please let us know.
>
>
> ###  **W1 : Connection between the proposed Spaced KD and the spacing effect in biological learning. Why does training for several additional epochs cause a spacing effect?**
>
> Our proposed Spaced KD for machine learning is inspired by the underlying computational principle of the spacing effect in biological learning:
>
> In machine learning, KD aims to optimize the parameters of a student network with the help of a teacher network by regularizing their outputs to be consistent in response to similar inputs. A pioneering theoretical analysis [1] shows that KD shares a similar mechanism with ensemble learning (EL) in improving generalization from the training set to the test set. In particular, online KD performs this mechanism at temporal scales, and self KD can be seen as a special case of online KD.
> In comparison, the biological spacing effect can also be generalized to a kind of EL at temporal scales, as the brain network processes similar inputs with a certain time interval and updates its synaptic weights based on previous synaptic weights, which allows for stronger learning performance at test time [2-4].
>
> The proposed Spaced KD draws inspiration from the biological spacing effect and capitalizes on the underlying connections between (online) KD and EL. It incorporates a space interval between the student network and the teacher network to improve the performance of generalization, validated through extensive empirical and theoretical analyses. We have added more explanations to make it clearer (see Sec.3.2, lines 193-207).
>
> > [1] Towards Understanding Ensemble, Knowledge Distillation and Self-Distillation in Deep Learning. ICLR (Outstanding Paper Nomination), 2023.
>
> > [2] The Phosphatase SHP2 Regulates the Spacing Effect for Long-Term Memory Induction. Cell, 2009.
>
> > [3] It’s All About Timing. Cell, 2009.
>
> > [4] The Right Time to Learn: Mechanisms and Optimization of Spaced Learning. Nature Reviews Neuroscience, 2016.
>
> ###  **W2 : Theoretical analysis of Spaced KD and its connection to offline KD.**
>
> Here we provide more explanations on our theoretical analysis of Spaced KD, as well as its connection to offline KD in ideal and practical conditions:
>
> The basic idea of our theoretical analysis lies in the discussion around the local optimum $\theta^*$ of the student network $f$, where the teacher network $g(\cdot, \phi)$ should also converge to $\phi^*$ although trained beforehand for a certain number of steps. With an ideal condition where $g$ and $f$ converge to the same local minima, offline KD and Spaced KD should perform identically best. However, this ideal condition hardly exists in practice, especially given the nature of over-parameterization in representative deep neural networks and the complexity of real-world data distributions. These two challenges result in a highly non-convex loss landscape of both $g$ and $f$ with a large number of local minima. Therefore, using a well-trained teacher in offline KD tends to be sub-optimal since $g$ and $f$ can easily converge to different local minima with SGD. In contrast, our Spaced KD introduces an appropriate space interval, which makes $g$ and $f$ somewhat distant but not too far apart. This design enjoys the theoretical benefits in terms of generalization while remaining effective in practice.
>
> To validate the above explanations, we present empirical results of offline KD and Spaced KD in the following table. It can be seen that our Spaced KD also slightly outperforms offline KD, and the performance lead tends to be more significant when using a wider network, suggesting the impact of over-parameterization on the mismatched assumption. Besides, the results of the critical period (see Fig.3) suggest that the benefit of Spaced KD is highly relevant to the later training stage (i.e., close to the convergence point), consistent with our theoretical analysis around $\theta^*$.
>
> | Student   | Teacher       | Offline KD | Spaced KD | Increase |
> | --------- | ------------- | ---------- | --------- | -------- |
> | ResNet-18 | ResNet-18 * 1 | 72.22      | 72.43     | **0.21** |
> | ResNet-18 | ResNet-18 * 2 | 72.53      | 72.77     | **0.24** |
> | ResNet-18 | ResNet-18 * 4 | 72.83      | 73.11     | **0.28** |
> | ResNet-18 | ResNet-18 * 8 | 73.04      | 73.52     | **0.48** |
> | ResNet-18 | DeiT-Tiny     | 78.80      | 79.38     | **0.58** |
> | ResNet-18 | PiT-Tiny      | 78.50      | 78.77     | **0.27** |
>
>
> Our theoretical and empirical analyses collectively demonstrate the particular role of Spaced KD in improving the generalization capability of the obtained solution. We have added these explanations in our revised manuscript (see lines 309-319 and Tab.8 in Appendix A.4).

---

> ### Author Response · Authors · 2024-11-19
> **Author Response (Part 2)**
>
> ### **W3: Comparison with other self KD methods.**
>
> As an easy-to-use and compatible strategy, the proposed Spaced KD can be combined with other self KD methods, such as DLB [5] and PSKD [6]. We reproduce all results with their official implementations and author-provided codes. As shown in the below table, our Spaced KD brings significant improvements to both self KD methods. We have added it to our revised manuscript (see lines 374-375 and Tab.4).
>
> Performance of self KD methods on ResNet-18 and CIFAR-100:
> | Method     | w/o KD | w/ KD  | w/ KD + Spaced KD |
> | ---------- | ------ | ------ | ----------------- |
> | DLB [5]    | 68.80  | 68.87  |     **69.31**     |
> | PSKD [6]   | 74.92  | 75.20  |     **75.38**     |
>
>
> > [5] Self-Distillation from the Last Mini-Batch for Consistency Regularization. CVPR, 2022.
>
> > [6] Self-Knowledge Distillation with Progressive Refinement of Targets. ICCV, 2021.
>
> ### **Q1: What are the numerical values of x and y in Fig.2a?**
>
> In Fig.2a, $x$ denotes inter-stimulus intervals between training sessions, and $y$ denotes the strength of long-lived memory trace. We have modified Fig.2a with numerical values of $x$ and $y$.
>
>
> ### **Q2 : As 0.5 epochs mean training on half of the dataset, would this cause a bias towards that half of the dataset? Is the training dataset shuffled?**
>
> As you understand it, 0.5 epochs mean training on half of the dataset. However, this would **not** cause a bias towards that half of the dataset.
> First, the training dataset is shuffled for each training epoch.
> Second, our specific implementation ensures that the teacher and the student receive the same data flow. For example, if $s = 0.5$, the teacher is trained on the first half of the dataset, and then the student is trained on the same first half of the dataset. After that, the teacher is trained on the second half of the dataset, and then the student is trained on the same second half of the dataset. But notice that the dataset is shuffled every time a new epoch starts, the specific images in half the dataset will also change with training iterations.
> Third, our proposed Spaced KD provides great improvements over a range of space intervals (i.e., 0.5, 1.0, 1.5, 2.0 epochs, see Fig.2b), further ruling out the possibility of dataset bias.
> We have added more explanations in our revised manuscript (see lines 220, 342-343).
>
>
> ### **Q3: How can the proposed method be applied to self-distillation in [7], where the teacher is the deepest layer of models and the students are the shallow layers?**
>
> In fact, the default implementation of ``self KD'' in our work is essentially identical to [7], i.e., the deepest layer (as the teacher) transfers knowledge to the shallow layers (as the student). To make this clearer, we have added more explanation (lines 346-350) and a pseudo code to Appendix A.9.
>
> > [7] Be Your Own Teacher: Improve the Performance of Convolutional Neural Networks via Self Distillation. ICCV, 2019.
>
> ### **Q4: What dataset was used for Fig.3, and which settings were used for the experiments?**
>
> We use CIFAR-100 in Fig.3 and $s=1.5$. In these experiments, we empirically investigate the impact of different initiating times of Spaced KD, which identifies a critical period of Spaced KD that is more effective at the later stage rather than the early stage of training. We have added more explanations to Fig.3.

---

> ### Author Response · Authors · 2024-11-24
>
> Dear Reviewer 5J6i,
>
> If you have any further feedback or concerns, please feel free to let us know. Your input is highly appreciated, and we look forward to hearing from you.
> Thank you once again for your time and consideration.
>
> Best wishes,
>
> Authors

---

> ### Comment · Reviewer_5J6i · 2024-11-24
> **Official Comments by Reviewer 5J6i**
>
> Thanks authors for their reply.
>
> W1: I still cannot see the connection between the proposed method (a learning process) and the biological spacing effect (which reduces forgetting of information). As mentioned in the authors' reply, "the brain network processes similar inputs with a certain time interval and updates its synaptic weights based on previous synaptic weights." How is this related to the proposed KD methods where teacher and student have different weights?
>
> W2: The loss functions for the teacher and student are different, which means they have different loss landscapes. How do they share the same local minima even in the ideal condition?
> Authors updated Section 4.2 (Line 309-311) and stated, "Spaced KD guides the student f with a well-defined trajectory established by the teacher g that is slightly ahead in training" I would like to highlight some related works [4] [5].
>
> W3: In PSKD, the teacher is one epoch before. Did the authors mean they used 1.5 epochs ahead when combining their method with PSKD?
> I cannot match the results in Tables 1, 2, and 4. For example, the results for ResNet18 on CIFAR-10 without KD are the same in Tables 1 and 2 but differ from the result for PSKD without KD in Table 4.
>
> [4] Follow Your Path: a Progressive Method for Knowledge Distillation.
>
> [5] Pro-KD: Progressive Distillation by Following the Footsteps of the Teacher

---

> ### Author Response · Authors · 2024-11-24
> **Comment on response of reviewer 5J6i**
>
> We appreciate your insightful comments and provide a point-to-point response below.
>
>
> ### **W1: Biological Connection**
>
> In general, spaced learning is a paradigm of the biological learning process, which improves the learning performance from the training condition to the similar testing condition, similar to the generalization from the training dataset to the test dataset in ANN, rather than directly handling the memory forgetting issue (as we described in the Sec.1).
>
> In line 204, we describe the biological learning process as "updates the synaptic weights based on previous synaptic weights" to set the stage for the discussion with ensemble learning from a temporal perspective, where the learner evolves its capability on the basis of its earlier state.
>
> Throughout the manuscript, we mainly focus on the online condition where the teacher and student have the same parameter space and initialization. In addition, the proposed KD is a regularization term for updating the student network by the teacher's output instead of constraining the parameters directly. So, it still works when extending to the KD methods where the teacher and student have different weights.
>
>
> ### **W2: Theoretical Analysis**
>
> As you correctly pointed out, the teacher and student models indeed have different loss landscapes, which raises the concern about how they can share the same minima. The "ideal condition" mentioned in lines 309-311 refers to the situation where the student model, initialized with the same parameters as the teacher and following the teacher's training trajectory, is able to converge to a well-trained minima akin to that achieved by offline KD. In this idealized case, the student can fully track the teacher’s learning path, allowing it to reach a similar final minima.
>
> However, as we discuss further after line 312, this ideal condition is rare and often difficult to achieve in practice, especially in dynamic training scenarios. Therefore, we introduce spaced intervals into the distillation process, which provides the student model with sufficient room to explore multiple convergence directions. This flexibility allows the student to find a better solution space, enhancing the distillation process and leading to performance improvements that often exceed traditional methods. The reason we introduce the "ideal condition" here is to set the stage for the subsequent comparison between the performance of offline KD and spaced KD. We will further refine this section of the explanation in the final draft.
>
> In line with the progressive learning paradigm, both ProKT [4] and Pro-KD [5] propose a similar learning framework to online KD, where the student model learns from an actively updating teacher during training. In ProKT, the teacher also receives a reverse distillation loss from the student, a concept we briefly mention in lines 158-160. We will cite these valuable works in the final draft to support our discussion.
>
> [4] Follow Your Path: a Progressive Method for Knowledge Distillation.
>
> [5] Pro-KD: Progressive Distillation by Following the Footsteps of the Teacher
>
> ### **W3: Comparision to other KD methods**
>
> As you kindly pointed out, the soft label from teachers in PSKD and DLB is one epoch earlier and one batch iteration earlier, respectively, where the teacher-student gap is pre-defined. Thus, we initiate a student network identical to the teacher, then we train the teacher model utilizing PSKD or DLB, and the student model is trained either online or in a spaced style with an interval of 1.5 epochs. Specifically, the results `w/o KD` of PSKD and DLB in Table 4 are the performance of the teacher model, `w/ KD` is the performance of online students, and `w/ Ours` corresponds to spaced students. It is worth mentioning that, we follow the exact training pipeline (including learning rate scheduler, optimizer, and dataset transformation, etc) of corresponding works when reproducing their results and incorporating our proposed Spaced KD framework, which is different from that of Table 1 and 2, and that explains the difference in numbers. We will add more details about the implementation of the comparative methods in our revised manuscript.
>
> Thank you again for your constructive feedback. We hope this response adequately addresses your concerns. Please feel free to reach out if you have any further questions or require additional clarification.

---

> > ### Author Response · Authors · 2024-11-30
> > **Looking forward to your feedback**
> >
> > Dear Reviewer 5J6i,
> >
> > We sincerely appreciate your valuable comments. We have tried our best to provide a point-to-point response and revise our manuscript accordingly. We hope that our extended experiments, explanations and analyses have addressed your concerns for more positive rating. Since the rebuttal period is coming to an end, we would appreciate further feedback from you. Please feel free to contact us if you have any further questions.
> >
> > Authors

---

> ### Comment · Reviewer_5J6i · 2024-12-03
> **reply of reviewer 5J6i**
>
> Thank you for your further reply.

---

### Official Review · Reviewer_Ac8Y · 2024-11-04

**Soundness:** 3
**Presentation:** 2
**Contribution:** 3
**Rating:** 5
**Confidence:** 4

**Summary:**

In this work, the authors perform a scheme on the lines of online and self KD to promote generalization of the model. They propose SpacedKD which is inspired from spacing effect in biological learning which highlights that study sessions across time is critical for memory formation. They argue that this spacing effect can be generalized to a kind of ensemble learning at the temporal scale. They implement this spaced KD in both online and self KD setups. For online KD, they just train the teacher in advanced for a certain number of steps termed as spaced interval. Then they just freeze this teacher and distill this knowledge to the student using the same data. They further provide a theoritical justification of its better generalization than online KD by analyzing the Hessian Matrix of the loss function of student in both the scenarios by arguing that if the student model converges to a local minimizer at step t of SGD then expected value of trace of hessian matrix using Spaced KD is lesser.  Furthermore, they provide experimental results to verify their claim on the CIFAR-100, ImageNet-100 and Tiny-ImageNet datasets for both self and online KD where they show significant gains.

**Strengths:**

The connection identified to the spacing effect in biological learning is quite interesting and provide intuitive backing for the method. The analysis of hessian of loss being lesser for spaced KD is also interesting and it being converging to a flatter local minima via SGD.
The results show prominent gains in performance which is again promising. Furthermore, the authors have also provided analysis of using different value of the interval various architectures along with the different architecture of teacher student pairs (inclduing different architectures for teacher). Also, the analysis in presence of various corruptions further backs the use case of their method.

**Weaknesses:**

Not exactly clear from the paper what is the exact difference between spaced and online KD setup. Also, why is the description of implementing spaced KD for self distillation not provided. I understand that it is pretty straightforward to adapt it but for completeness it should be there and there might be some smal changes that would be required into a naive adaptation that helps improving the result.

Regarding results, even though the authors have mentioned in one of the figure captions the reason for not providing results on ViT architecture, I feel that it is not justified. To test its applicability it should be tested on state of the architectures and large scale datasets. The major results revolve around CIFAR-100, Tiny-ImageNet and ImageNet-100 architectures and only one result is provided in the ImageNet-1k (and that too with a ResNet-18 architecture) leading to speculations that it might not give this sort of gains on ImageNet-1k or other large scale settings. Also, I dont understand what the Appendix A.5 results want to convey.

**Questions:**

Please check the weakness section.  Appendix A.5 is confusing. Does it imply that distilling from a teacher trained s steps ahead is not useful? But then this is the method right that first you should update teacher for s steps and then using that distill to student for those s steps. Also it would have been very useful to provide a psuedo code comparison for online and spaced KD so as to what exactly is the differennce. Overall I feel the presentation of the paper could have been improved especially the emipirical section.

Anyhow, right now I am going with the marginally below acceptance but I would be happy to reconsider my rating based on the justifications provided in the rebuttal.

---

> ### Author Response · Authors · 2024-11-19
> **Author Response**
>
> Thank you for your valuable comments. Below, we provide a point-to-point response to each of the weaknesses (**W**) and questions (**Q**), and summarize the corresponding revisions. We hope you may consider this a sufficient reason to raise the score. If you have any further questions, please let us know.
>
> ### **W1: Implementation of online KD, self KD and its spaced version.**
>
> In standard online KD, the teacher model transfers knowledge to the student model at each training time step. In our spaced version, we first train the teacher model for $s$ steps ahead, and then train the student model for $s$ steps. This two-step training process alternates between the two models. For example, if $ s = 1.5 $ epochs, the teacher model is updated using cross-entropy loss for 1.5 epoch, and then the student model is updated using the KD loss for 1.5 epoch.
>
> In self KD, the deepest layer (as the teacher) transfers knowledge to the shallow layers (as the student) at each training time step [1]. In our spaced version, we first train the model using the cross-entropy loss between the deepest layer's output and the ground-truth label for $s$ steps. We then train the model using the standard self KD loss [1] between the deepest layer's output and each shallow layer's output for $s$ steps.
>
> We have added more explanations to make it clearer (see lines 178-186, lines 214-215, lines 346-350). We further provide a pseudo-code of online KD, self KD, and its spaced version in Appendix A.9.
>
> > [1] Be Your Own Teacher: Improve the Performance of Convolutional Neural Networks via Self Distillation. ICCV, 2019.
>
>
> ### **W2: Advanced architectures and large-scale datasets.**
>
> In our original submission, we mainly focus on online KD, evaluating it and its spaced version across a variety of architectures (ResNet and Transformer) and datasets (CIFAR-100, Tiny-ImageNet and ImageNet-100/1k). As analyzed in Sec.4.2, we treat self KD as a special case of online KD and perform validation experiments using ResNet-based architectures on CIFAR-100, Tiny-ImageNet and ImageNet-100. Following your suggestion, we evaluate self KD and its spaced version with ViT and ImageNet-1k.
>
> Results of self KD using transformer-based architectures:
>
> | Dataset       | Network   | w/o KD | Self KD | self KD w/ ours |
> | ------------- | --------- | ------ | ------- | --------------- |
> | CIFAR-100     | DeiT-Tiny | 64.77  | 65.24   | **68.26**       |
> | Tiny-ImageNet | DeiT-Tiny | 50.17  | 49.73   | **53.59**       |
> | ImageNet-100  | DeiT-Tiny | 69.52  | 70.50   | **73.46**       |
>
> Results of self KD using ResNet-18 on ImageNet:
>
> | Epochs           |   20   |   40   |   60   |   80   |
> | ---------------- | ------ | ------ | ------ | ------ |
> | self KD w/o ours |  44.87 |  46.78 |  47.78 |  47.99 |
> | self KD w/ ours  |  47.03 |  49.94 |  50.22 |  51.57 |
>
>
> Due to the limited time to prepare the rebuttal, the experiments of self KD using ViT on ImageNet-1k are still under training. Based on the current training curve, our proposal is expected to largely improve self KD in this large-scale setting. We hope to finish it before the discussion period and will add it to our revised manuscript.
>
>
> ### **W3 & Q1: Explanation of Appendix A.5.**
>
> In Appendix A.5 we consider a naive baseline of implementing the proposed spacing effect. Specifically, we train the teacher model for $s$ steps before transferring knowledge to the student model at each step during the following training time. In other words, the teacher model keeps constant $s$ steps ahead of the student model. However, such a naive baseline exhibits no significant improvement over online KD, consistent with our empirical analysis (see Fig.3) and theoretical analysis (see Sec.4.2): The teacher model of Spaced KD can provide a stable informative direction for optimizing the student model after each $s$ steps, whereas the teacher model of the naive baseline fails in this purpose due to its ongoing changes when optimizing the student model. Such different effects also suggest that the implementation of the spacing effect is highly non-trivial and requires specialized design as in our work.
> We have added more explanations to Appendix A.5 to make it clearer.

---

> ### Author Response · Authors · 2024-11-24
>
> Dear Reviewer Ac8Y,
>
> If you have any further feedback or concerns, please feel free to let us know. Your input is highly appreciated, and we look forward to hearing from you.
> Thank you once again for your time and consideration.
>
> Best wishes,
>
> Authors

---

> ### Author Response · Authors · 2024-11-24
> **Author Response (Part 2)**
>
> ### **W2 (part 2): Advanced architectures and large-scale datasets.**
>
> We have finished experiments on self KD and online KD using ViT on ImageNet-1k, and the results are summarized in the table below. These experiments further validate the effectiveness of our proposed method in improving performance on large-scale datasets over standard online KD and self-KD.
>
> | Epoch             | 100       | 200       | 300       |
> | ----------------- | --------- | --------- | --------- |
> | w/o KD            | 58.31     | 66.14     | 72.29     |
> | online KD         | 58.18     | 65.93     | 72.04     |
> | online KD w/ ours | **58.47** | **66.54** | **72.34** |
> | self KD           | 58.81     | 66.37     | 72.39     |
> | self KD w/ ours   | **60.82** | **67.27** | **73.69** |
>
> These results, which we have finalized since the initial submission, demonstrate significant improvements in performance across the training epochs when applying our method. We believe these additional experiments further strengthen the validity of our proposed approach for large-scale datasets like ImageNet-1k.
>
> If you have any further questions or concerns, please feel free to let us know at any time.

---

> > ### Comment · Reviewer_Ac8Y · 2024-11-28
> >
> > Thanks for addressing the concerns I mentioned. However, the VIT results on Imagenet-1k are not very promising and the initial speculation of this not scaling up. I think more investigation on its effectiveness in a large scale setup is needed.

---

> > > ### Author Response · Authors · 2024-11-29
> > > **Response to Reviewer Ac8Y**
> > >
> > > Thank you for your valuable feedback.
> > >
> > > Our work aims to improve generalization performance of KD without extra data and parameters. We therefore select online KD and self KD as the baseline approaches. Due to the vary limited time and computational resources during the rebuttal period, we can only test a specific configuration of both approaches, which remain sub-optimal without further adjustments. Moreover, since the setup of online KD in ViT remains largely under explored in literature, a direct implementation of this baseline approach results in even negative impact.
> > >
> > > Under the specific configuration, online KD and self KD outperform w/o KD by only **-0.25%** and **0.10%**. In contrast, our Spaced KD outperforms online KD and self KD by **0.30%** and **1.30%**. Given that the results were obtained without extra data, parameters or hyperparameter tuning, we believe that the improvements of Spaced KD are comparably pronounced. The robustness of Spaced KD also make it more applicable in real world scenarios (although performance may not be tuned to be optimal). Furthermore, in ResNet-18/ImageNet-1K, our spaced version of online KD and self KD outperform w/ours by **5.08%** and  **3.58%**, demonstrating its clear superiority with relatively large-scale dataset.
> > >
> > > We sincerely appreciate your insightful comments, and hope you may understand the very limited time and resource for tuning large-scale experiments in rebuttal. We will provide a more extensive analysis for the implementation of online KD and self KD as well as their spaced versions in ViT/ImageNet-1K, and will add them in the final version. We would appreciate it if you could take the sub-optimality of baseline KD approaches and the robustness of our Spaced KD in to account and reconsider the rating. Please feel free to contact us if you have any further questions.

---

> > > > ### Author Response · Authors · 2024-11-30
> > > > **Looking forward to your feedback**
> > > >
> > > > Dear Reviewer Ac8Y,
> > > >
> > > > We sincerely appreciate your valuable comments. We have tried our best to provide a point-to-point response and revise our manuscript accordingly. We hope that our extended experiments, explanations and analyses have addressed your concerns for more positive rating. Since the rebuttal period is coming to an end, we would appreciate further feedback from you. Please feel free to contact us if you have any further questions.
> > > >
> > > > Authors

---

### Author Response · Authors · 2024-11-19
**Overall Response**

Dear reviewers,

We sincerely appreciate your great efforts in reviewing our manuscript and all your valuable feedback. It is encouraging to see many positive comments, especially noting that our method is "interesting" (Reviewers Ac8Y, bFLi), "easy to follow" (Reviewer 5J6i, bFLi, v8iU) and "effective" (Reviewers 5J6i, bFLi, v8iU). We have provided a point-to-point response to all comments, summaized the corresponding changes, and uploaded a revised version of our manuscript. Our responses and revisions can be summarized as follows:

(1) We have added more explanations to the connections between the proposed Spaced KD and biological spacing effect.

(2) We have added more explanations to the theoretical analysis of offline KD, online KD and Spaced KD, demonstrating the superiority of our proposal.

(3) We have added more explanations to the implementation of online KD, self KD and their spaced version, with the coressonding pseudo code.

(4) We have added more in-depth and extensive experiments across a variety of network architectures and benchmark datasets, and the proposed Spaced KD achieves consistent improvements.

We believe that the quality of the current manuscript has been largely improved. We hope you may consider this a sufficient reason to raise the score. If you have any further questions, please let us know.

---

> ### Author Response · Authors · 2024-11-23
> **Looking forward to your feedback**
>
> Dear reviewers,
>
> We sincerely appreciate your great efforts in reviewing our manuscript and all your valuable feedback. We are pleased to see many positive comments, and have tried our best to address the remaining concerns. We know and understand that the reviewers are very busy, and sincerely hope you could take a little time to review our responses, while let us know whether you have further questions. We are willing to try our best to answer these questions. Great thanks!
>
> Best wishes,
>
> Authors

---

### Comment · Area_Chair_QFvE · 2024-11-26
**Encouragement to Actively Participate in the Discussion Phase**

Dear Reviewers,

Thank you for your valuable contributions to the review process so far. As we enter the discussion phase, I encourage you to actively engage with the authors and your fellow reviewers. This is a critical opportunity to clarify any open questions, address potential misunderstandings, and ensure that all perspectives are thoroughly considered.

Your thoughtful input during this stage is greatly appreciated and is essential for maintaining the rigor and fairness of the review process.

Thank you for your efforts and dedication.

---

### Meta-Review · Area_Chair_QFvE · 2024-12-20

**Metareview:**

(a) Summary of Scientific Claims and Findings
This paper introduces Spaced Knowledge Distillation (Spaced KD), a novel approach inspired by the biological "spacing effect." The proposed method introduces intervals between teacher and student training updates to enhance the generalization performance of knowledge distillation (KD). Key contributions include:
A spacing interval mechanism applied to online and self-distillation settings.
A theoretical justification using Hessian trace analysis to link the method to flatter minima, supporting better generalization.
Empirical validation on multiple datasets and architectures, showing performance gains over traditional KD methods.

(b) Strengths of the Paper
Interesting Bio-inspiration: The application of the spacing effect to KD is innovative and provides an intuitive motivation for the method.
Comprehensive Empirical Analysis: The paper includes extensive experiments across CIFAR-100, Tiny-ImageNet, ImageNet-100, and ImageNet-1k with various architectures, demonstrating performance improvements.
Theoretical Contributions: The authors provide a theoretical explanation of the mechanism using Hessian trace analysis, linking Spaced KD to flatter loss landscapes.
Practical Simplicity: Spaced KD is easy to implement, requiring minimal changes to existing KD pipelines.

(c) Weaknesses of the Paper
Incremental Contribution: The method primarily adapts an existing concept (spacing effect) without significantly advancing the theoretical or methodological understanding of KD.
Limited Scalability Validation: The performance gains on large-scale datasets (e.g., ImageNet-1k) and transformer-based architectures (e.g., ViT) were marginal and less compelling.
Unconvincing Biological Connection: The claimed link between the biological spacing effect and the proposed KD method is weakly justified and does not align with the mechanisms observed in biological learning.
Inconsistent Results: Improvements over state-of-the-art methods like PSKD and DLB are minimal, and the lack of adaptive interval tuning limits generalizability across diverse settings.
Weak Baseline Comparisons: The paper fails to benchmark against more recent or advanced KD methods and does not adequately explore scenarios involving different teacher-student architectures.

(d) Reasons for Rejection
Lack of Substantial Novelty: While the bio-inspiration is interesting, the contribution is primarily incremental, adapting a known concept without introducing significant theoretical or methodological advancements.
Limited Impact on Large-Scale Settings: The method's limited effectiveness on larger datasets and advanced architectures undermines its broader applicability and scalability.
Weak Empirical Justifications: The minimal improvements over existing KD methods and lack of robust comparisons with state-of-the-art baselines weaken the empirical claims.
Unresolved Reviewer Concerns: Despite the authors’ efforts during the rebuttal, key issues such as the biological connection, baseline completeness, and scalability remained inadequately addressed.

**Additional Comments On Reviewer Discussion:**

Points Raised by Reviewers and Author Responses

Concern: Reviewers questioned the novelty of the work, noting that the application of the spacing effect to KD adapts an existing concept rather than introducing a fundamentally new idea.
Author Response: The authors argued that the spacing effect has not been explored in KD and provided additional analyses to justify its relevance. They emphasized the practical simplicity and ease of integration into existing KD frameworks.
Evaluation: While the clarification highlighted the method’s practicality, the reviewers remained unconvinced about its novelty and broader impact.

Concern: Reviewers expressed skepticism about the weak connection between the biological spacing effect and the proposed KD method, as the biological mechanisms are not explicitly modeled.
Author Response: The authors acknowledged the simplified biological analogy but argued that the spacing mechanism’s empirical benefits validate the inspiration.
Evaluation: The reviewers appreciated the acknowledgment but maintained that the biological justification was overstated and not central to the method’s effectiveness.

Concern: The paper lacked benchmarks against recent and advanced KD methods and failed to explore a variety of teacher-student architecture combinations.
Author Response: The authors added comparisons with a few additional methods and conducted experiments with different architectures, but the new results showed marginal improvements.
Evaluation: The additional comparisons were noted but did not convincingly address the concern, as the results failed to demonstrate significant gains over stronger baselines.

Concern: Reviewers raised concerns about the method’s marginal improvements on large-scale datasets (e.g., ImageNet-1k) and its limited validation on transformer-based architectures.
Author Response: The authors conducted additional experiments on larger datasets and ViT-based models but found that the gains were inconsistent. They attributed this to task-specific limitations of the spacing mechanism.
Evaluation: The reviewers found this explanation reasonable but ultimately felt that the inconsistent results undermined the method’s scalability and generalizability.

Despite the authors’ efforts during the rebuttal period to address concerns with additional experiments, comparisons, and analyses, key issues remained unresolved. The method’s contribution is incremental, with limited novelty and marginal improvements over existing KD approaches. Scalability and generalizability are constrained by inconsistent performance on large datasets and transformer-based architectures, while the weak biological justification further detracts from the conceptual appeal. Additionally, the lack of adaptive interval tuning and incomplete baseline comparisons reduce the method’s practical relevance.

---

### Decision · Program_Chairs · 2025-01-22

Reject